# The Immunomodulatory Role of Vitamin D in Regulating the Th17/Treg Balance and Epithelial–Mesenchymal Transition: A Hypothesis for Gallbladder Cancer

**DOI:** 10.3390/nu16234134

**Published:** 2024-11-29

**Authors:** Ricardo Cartes-Velásquez, Agustín Vera, Rodrigo Torres-Quevedo, Jorge Medrano-Díaz, Andy Pérez, Camila Muñoz, Hernán Carrillo-Bestagno, Estefanía Nova-Lamperti

**Affiliations:** 1School of Medicine, University of Concepcion, Concepcion 4070409, Chile; rtorresquevedo@yahoo.es (R.T.-Q.); hecarrillo@udec.cl (H.C.-B.); 2Molecular and Translational Immunology Laboratory, Department of Clinical Biochemistry and Immunology, Pharmacy Faculty, University of Concepcion, Concepcion 4070409, Chile; agvera2016@udec.cl (A.V.); camilapmunoz@udec.cl (C.M.); 3Hepatopancreatobiliary Surgical Unit, Service of Surgery, Hospital Guillermo Grant Benavente, Concepcion 4070022, Chile; 4Hepatopancreatobiliary Surgical Unit, Service of Surgery, Hospital Las Higueras, Talcahuano 4270918, Chile; jmedranod@gmail.com; 5Department of Instrumental Analysis, Pharmacy Faculty, University of Concepcion, Concepcion 4070409, Chile; aperezd@udec.cl; 6Facultad de Odontología y Ciencias de la Rehabilitación, Universidad San Sebastián, Concepción 4080871, Chile; 7Service of Medicine, Hospital Las Higueras, Talcahuano 4270918, Chile

**Keywords:** cancer, vitamin D, T cell, epithelial–mesenchymal transition

## Abstract

The etiology of gallbladder cancer (GBC) is multifactorial, with chronic inflammation resulting from infections, autoimmune diseases, and lifestyle factors playing a pivotal role. Vitamin D deficiency (VDD) has been implicated in the pathogenesis of autoimmune disorders and various malignancies, including GBC. Research on autoimmune diseases highlights the anti-inflammatory properties of vitamin D, suggesting its potential to mitigate disease progression. In oncology, VDD has similarly been linked to increased inflammation, which may contribute to both the initiation and progression of cancer. A critical component in carcinogenesis, as well as in the immunomodulatory effects of vitamin D in autoimmune conditions, is the balance between T-helper 17 (Th17) cells and regulatory T (Treg) cells. We hypothesize that vitamin D may inhibit epithelial–mesenchymal transition (EMT) in GBC by modulating the spatial distribution of tumor-infiltrating T cells, particularly through the regulation of the Th17/Treg balance at the tumor margins. This Th17/Treg imbalance may act as a mechanistic link between VDD and the progression of GBC carcinogenesis. Investigating the role of an Th17/Treg imbalance as a mediator in VDD-induced EMT in GBC not only provides deeper insights into the pathogenesis of GBC but also sheds light on broader mechanisms relevant to the development of other solid organ cancers, given the expanding recognition of the roles of VDD and Th17/Treg cells in cancer biology.

## 1. Introduction

The primary role of the gallbladder is to store and concentrate bile, an alkaline fluid produced by the liver that helps digest and absorb lipids, which do not dissolve in water. Bile is composed of cholesterol, bilirubin, bile salts, phospholipids, and water. When fatty acids and amino acids enter the duodenum, specialized I-cells release cholecystokinin (CCK), which prompts the gallbladder to contract and release bile while relaxing the sphincter of Oddi, permitting bile to flow into the duodenum. Bile salts, derived from cholesterol, play a vital role in emulsifying lipids and forming micelles that enhance lipid digestion. Bile acids (BAs) function as ligands for several receptors, including the farnesoid X receptor (FXR), the vitamin D receptor (VDR), and the G protein-coupled receptor (TGR5), thereby impacting metabolic balance and immune responses [1,2,3].

Gallbladder cancer (GBC) ranks 20th in terms of its incidence rate and 17th regarding its death rate [4]. Projections indicate a substantial increase of more than 75% in the global burden of GBC by 2040 [5]. The clinical challenge lies in the silent onset and rapid metastasis of GBC, leading to late-stage diagnoses and poor prognoses [5,6]. A considerable proportion of GBC cases, approximately 50%, are fortuitous discoveries during surgery or postoperative pathological analyses following a routine cholecystectomy [7]. However, routine biopsies often miss dysplasia changes, complicating accurate disease staging [8]. Despite recent advances in identifying potential biomarkers for GBC diagnosis and prognosis based on molecular pathways, their translation into clinical practice remains limited [9]. Notably, the neutrophil-to-lymphocyte ratio (NLR) stands as the most validated and widely utilized prognostic score, yet early diagnosis remains challenging [10]. While chemotherapy and emerging immunotherapy approaches are promising therapeutic strategies, surgical resection remains the best option for GBC, particularly in the early stages [11].

Vitamin D plays a well-established role in maintaining calcium phosphate homeostasis and regulating bone turnover throughout life. Its deficiency is associated with increased bone turnover, decreased bone density, and an elevated risk of fractures [12]. There are two forms of vitamin D: D2 or ergocalciferol, which is produced by ultraviolet (UV) irradiation of ergosterol, a sterol in fungi; and vitamin D3 or cholecalciferol, which is synthesized in animals. Vitamin D3 is produced when the precursor 7-dehydrocholesterol is exposed to UV light or obtained through the consumption of oily fish. Subsequently, it undergoes hydroxylation in the liver to form 25(OH)D3 and further hydroxylation in the kidney to generate its active form, 1,25(OH)D3 [13]. Vitamin D binds to VDR, which may be found in the cytoplasm, nucleus, or both. As a member of the nuclear receptor superfamily, VDR shares significant structural and ligand-binding similarities across various species. When a ligand binds, it triggers a conformational change in the AF-2 region, allowing the release of accessory proteins, exposing the DNA-binding pocket, and recruiting coactivators. The vitamin D/VDR complex regulates gene transcription by heterodimerizing with retinoid X receptor (RXR) isoforms and binding to vitamin D-responsive elements (VDREs) in the promoter regions of target genes, which can vary in structure. Despite its strong, non-covalent binding to VDR, vitamin D can also prompt rapid responses in target cells, such as calcium flux, the activation of secondary messenger systems, and the stimulation of cytosolic kinases [14]. In addition to vitamin D, bile acids, such as lithocholic acid, can also act as selective modulators of VDR. Furthermore, VDR activation by vitamin D influences bile acid metabolism, highlighting the complex interactions at this level. 

The association between UV light exposure, vitamin D deficiency (VDD, serum 25-hydroxyvitamin D [25(OH)D] < 50 nmol/L or 20 ng/mL) [15], and cancer has been extensively investigated since 1980, with observational and ecological studies providing substantial support [16]. Despite the detrimental impact of VDD, its prevalence globally ranges from 4% to 18%, with VDD affecting approximately 24% to 49% of the population [17]. Notably, vitamin D production by the skin appears to decrease the risk of various solid cancers, including those affecting the gallbladder, stomach, colorectal system, liver, pancreas, lung, breast, prostate, bladder, and kidney [18]. Furthermore, VDD has been linked to adverse outcomes in patients with GBC [19]. Recent systematic reviews have corroborated the association between VDD or vitamin D intake and various cancers [20,21]. It is noteworthy that VDD is particularly prevalent in countries with high GBC rates, highlighting this concerning association [17].

While the precise biological mechanisms underpinning the relationship between VDD and cancer remain incompletely understood, the evidence suggests direct effects of vitamin D on cancer cells through anti-inflammatory, anti-angiogenic, and anti-proliferative mechanisms [18,19]. Additionally, emerging research indicates a potential impact of vitamin D on the immune response against cancer, albeit with conflicting findings. Some studies propose an enhancement of specific immune response aspects, such as the inhibition of apoptosis induced by inflammatory cells and tumor growth, while others highlight impairment, including compromised T cell effector functions and the induction of a tolerogenic profile [20,21]. Thus, further investigation is warranted to elucidate the intricate immunological mechanisms linking VDD, immune response, and cancer.

The balance between T-helper 17 (Th17) and regulatory T (Treg) cells is crucial for maintaining immune homeostasis. The disruption of this balance—whether through Th17 dominance or impaired Treg function—has been associated with autoimmune disorders. Both Th17 and Treg cells, along with their associated cytokines, have been implicated in tumorigenesis, with the ability to either promote or inhibit tumor development. However, the precise mechanisms that govern these opposing roles remain poorly understood [22].

Epithelial–mesenchymal transition (EMT) is a biological process characterized by the loss of epithelial cell polarity and intercellular adhesion, leading to the acquisition of a mesenchymal phenotype. During EMT, there is a downregulation of epithelial markers such as E-cadherin, β-catenin, desmoplakin, cytokeratins, and laminin, accompanied by an upregulation of mesenchymal markers such as *N-*cadherin and vimentin [23]. EMT is considered a critical step in the development of invasiveness and metastasis in various epithelial cancers, including GBC, and multiple molecular mechanisms contribute to EMT [24,25]. Interestingly, EMT is associated with VDD [26], which appears to be an immunological mechanism linking VDD and GBC.

This paper aims to critically evaluate the current literature on the influence of VDD on GBC, focusing on the immunological pathways implicated in its carcinogenesis, drawing insights from cancer biology and autoimmune research.

## 2. Gallbladder Cancer Risk Factors

Numerous environmental factors are implicated in the development of GBC, though the evidence remains inconclusive for some. A recent systematic review evaluated 215 primary studies and conducted 350 meta-analyses across seven domains, identifying strong associations between increased GBC risk and factors such as body mass index (BMI), hip and waist circumference, obesity, and bile duct infections, while higher education was linked to a reduced risk. The credibility of 39 other associations, including those related to smoking, alcohol, and physical inactivity, was lower. The study emphasizes the need for further high-quality research to confirm these findings and guide prevention efforts in high-incidence regions [27].

Gallstones are a major risk factor, present in 70–98% of cases, though only a small percentage of individuals with gallstones develop GBC. Larger gallstones, particularly cholesterol stones, which are prevalent in high-risk ethnic groups, are believed to contribute to carcinogenesis through persistent mucosal irritation and chronic inflammation. This inflammation can lead to DNA damage, repeated cycles of tissue repair, and cytokine release, which together promote karyokinetic changes [28,29]. Chronic cholecystitis, linked to gallstones and recurrent trauma, is thought to drive neoplasia over time. Additionally, abnormal pancreaticobiliary duct junctions have been associated with an increased risk of biliary and pancreatic cancers, particularly in patients without bile duct cysts. Infection-related factors, including chronic *Salmonella* typhi carriers and *Helicobacter bilis* colonization, have also been connected to an increased risk of GBC, particularly in regions endemic with Salmonella [5,28,29].

Geographically, GBC incidence varies significantly, with higher rates reported in Latin America [30,31,32] and Asia, and lower rates observed in the United States and Western Europe. Indigenous populations are disproportionately affected, as global cancer registry data indicate, with northern India, Pakistan, and Korea displaying notably higher rates, underscoring ethnic disparities across regions [29].

Exposure to heavy metals such as nickel and cadmium has been suggested as a potential risk factor, although definitive associations are still lacking. Individuals in occupations like mining, where radon exposure is common, alongside tobacco users, are at an increased risk for both lung and gallbladder cancer. Certain pharmaceuticals, including methyldopa and isoniazid, have also been linked to an elevated gallbladder cancer risk [29].

Oncogenic factors such as nuclear factor kappa B, reactive oxygen species, inflammatory cytokines, prostaglandins, and micro-RNAs may further influence processes like cell proliferation, apoptosis, DNA mutation, and angiogenesis [5,29]. GBC typically shares histological characteristics with other biliary tract cancers (BTCs), as more than 90% of these cancers are adenocarcinomas. However, genomic profiling has uncovered distinct molecular alterations that differentiate GBC from other BTCs. While BTCs commonly exhibit mutations in pathways such as FGF, IDH, and PI3KCA, GBC is notably characterized by HER2 alterations. Mutations in TP53 and KRAS are prevalent across all BTCs [5,33,34,35,36]. Additionally, genetic polymorphisms associated with increased GBC susceptibility, including variants in toll-like receptors (TLR2 and TLR4), cytochrome P450 1A1 (CYP1A1), and the ATP-binding cassette transporter ABCG8, have emerged as potential biomarkers. These biomarkers could be instrumental in identifying high-risk patients who might benefit from early interventions such as cholecystectomy, especially given the challenges in diagnosing GBC due to the absence of specific early symptoms. Advancements in our understanding of the molecular mechanisms underlying GBC have highlighted various cellular events, such as HER2 aberrations, a high tumor mutational burden, and microsatellite instability, that could influence patient management. Despite the promising nature of these markers, there remains a significant need for reliable and validated biomarkers to enhance the clinical management and decision-making processes for GBC patients [37]. Genetic mutations in GBC can be either germline or somatic, involving key proto-oncogenes, tumor suppressor genes, and pathways like MAPK and PI3K, which regulate cell proliferation and apoptosis. Although similar mutations in other cancers, such as ALK, EGFR, and KRAS in lung cancer or BRAF in melanoma, have revolutionized cancer treatment, identifying analogous markers in GBC is challenging due to its genetic heterogeneity. Furthermore, the methylation status of oncogenes and tumor suppressor genes, which varies across different regions, also plays a crucial role in GBC carcinogenesis, with genes like PTEN, APC, P16, and MLH1 showing region-specific methylation patterns [38].

Despite the significance of individual risk factors for GBC, the interplay between genetic, lifestyle, and sociodemographic factors must be acknowledged. Feroz et al. [39] highlight that the GSTT1 (null) genotype significantly increases GBC susceptibility in the North Indian population. This genotype is particularly prevalent among female patients, those with adenocarcinoma, and individuals living in rural areas, suggesting a heightened risk of GBC development in these groups. Additionally, the combination of GSTT1 (non-null)/GSTP1 (Ile/Val + Val/Val) genotypes is associated with increased GBC risk in North Indians. Feroz et al. [40] further reports that overall GBC survival in the Gangetic belt is poor, with prognosis being influenced by geographic region, the presence of gallstones, and tumor stage, while surgery serves as a protective factor. In Chile, Boektegers et al. [32] developed two risk-scoring systems to predict GBC, one of which includes genetic factors. This system, which considers the proportion of Mapuche (largest indigenous populations in Chile) ancestry and the rs17209837 genotype, improves risk prediction and reduces the number of cholecystectomies needed to prevent a single GBC case by 12. Zollner et al. [31] found a causal relationship between Mapuche ancestry and increased GBC risk, as well as gallstone disease, suggesting that gallstones may mediate the relationship between Mapuche ancestry and GBC. Interestingly, a higher proportion of Mapuche ancestry is inversely related to BMI. These findings have significant implications for GBC prevention, particularly in populations with high levels of Mapuche ancestry, and underscore the potential of incorporating genetic ancestry into primary and secondary prevention strategies, especially since the observed association between Mapuche ancestry and GBC risk is likely free from confounding factors.

The evidence underscores the importance of integrating genetic factors into GBC risk assessment, particularly in high-risk populations, where specific genotypes and genetic ancestry significantly influence susceptibility. The findings also highlight the potential benefits of personalized prevention strategies that incorporate genetic profiling, which could enhance the accuracy of risk prediction and reduce unnecessary interventions, despite the associated costs. Lastly, the observed relationships between genetic factors, gallstone disease, and GBC risk emphasize the need for further research into the underlying mechanisms and the development of targeted prevention and treatment approaches.

## 3. Vitamin D Deficiency/Supplementation and Inflammation

Beyond its well-known functions in bone health, vitamin D has been increasingly recognized for its immunomodulatory roles in recent decades. This is primarily mediated through the presence of a vitamin D receptor (VDR) in most tissues, including immune cells [12]. Extensive evidence has linked VDD with the development of autoimmune diseases (AIDs) such as systemic lupus erythematosus [41], thyrotoxicosis [42], type 1 diabetes [43], multiple sclerosis [44], Crohn’s disease and ulcerative colitis [45], seropositive rheumatoid arthritis [46], and polymyalgia rheumatica [47]. Numerous studies have demonstrated the potential of vitamin D supplementation in preventing or attenuating the severity of infections and AIDs [48]. In colorectal cancer (CRC) patients, vitamin D deficiency has been linked to higher post-operative *C-*reactive protein (CRP) levels and poorer overall survival rates, regardless of age. Correcting VDD may mitigate chronic inflammation and improve prognosis in these patients [49,50]. However, the optimal dosage of vitamin D remains uncertain, likely being influenced by genetic variations among individuals [51].

Obesity is associated with elevated levels of various inflammatory biomarkers and inversely correlated with 25-hydroxyvitamin D (25(OH)D) levels. Although there is a causal association between BMI and certain inflammatory biomarkers, observational and Mendelian randomization studies have not shown a mediating effect of 25(OH)D on this association [52]. But other non-linear Mendelian randomization analyses showed an L-shaped association between genetically predicted serum 25(OH)D and serum CRP, where CRP levels decreased sharply with increasing 25(OH)D concentration within the deficiency range (<25 nmol/L) and leveled off at ~50 nmol/L of 25(OH)D. The association between 25(OH)D and CRP is likely caused by VDD, suggesting that correcting low vitamin D status may reduce chronic inflammation [53].

Vitamin D supplementation (VDS) has shown significant reductions in serum tumor necrosis factor (TNF)-α levels, which may be particularly beneficial for patients with cancer or precancerous lesions by potentially suppressing tumor-promoting inflammatory responses [54]. Additionally, combined supplementation of vitamin D3 and omega-3 fatty acids has demonstrated decreased levels of CRP and TNF-α, along with a significant reduction in interleukin (IL)-6 levels in CRC patients. This combined intervention also exhibited positive effects on nutritional status, including increased weight, BMI, and fat-free mass percentage [55]. In cancer patients, particularly those with pancreatic ductal adenocarcinoma (PDAC) and CRC, low vitamin D levels are associated with elevated levels of pro-inflammatory cytokines such as IL-6, CRP, and TNF-α. In breast cancer survivors, VDS has shown potential in modulating inflammatory biomarkers, particularly TNF-α levels, depending on VDR single-nucleotide polymorphisms (SNPs) and haplotypes [56].

The evidence regarding long-term VDS presents mixed outcomes. For postmenopausal women, prolonged use of calcium and vitamin D supplements has been associated with reduced cancer mortality and increased cardiovascular disease (CVD) mortality after over 20 years of follow-up, with no significant impact on all-cause mortality [57]. In adults aged 50 years or older without clinically significant depressive symptoms at baseline, VDS did not yield statistically significant differences in the incidence or recurrence of depression or mood score changes over a median follow-up of 5.3 years, suggesting that vitamin D3 may not be effective in preventing depression [58]. In contrast, long-term (30-month) VDS was found to significantly reduce fasting insulin levels in the Homeostatic Model Assessment for Insulin Resistance (HOMA-IR), and serum concentrations of non-HDL cholesterol, hs-CRP, and uric acid in middle-aged to elderly patients with type 2 diabetes (T2D), though the effects might be influenced by vitamin D status, gender, and baseline obesity [59]. Although, high-dose monthly VDS in older adults did not prevent antibiotic use but was associated with fewer days on antibiotics per person-year [60]. Among adults with T2D, neither VDS nor *n*-3 fatty acids supplementation resulted in reductions in IL-6, hsCRP, or *N*-terminal pro–B-type natriuretic peptide (NT-proBNP) over a five-year period [61].

In children, however, the effects of VDS differ. Higher-than-standard VDS during the first two years of life was linked to a reduced risk of internalizing problems (symptoms of depression, anxiety, and somatization) between ages 6 and 8 years [62]. For school-aged children with initially low vitamin D levels, a weekly dose of 14,000 IU of oral vitamin D3 for three years effectively raised 25(OH)D concentrations but did not affect growth, body composition, or pubertal development [63]. Additionally, VDS did not contribute to weight reduction in children and adolescents with vitamin D insufficiency who were undergoing a weight management program [64].

Moreover, an epidemiological association has been noted between VDD and susceptibility to COVID-19, along with the progression of severe forms of the disease indicating its potential utility as a component to safeguard SARS-CoV-2 patients, accelerate recovery, mitigate symptom severity, and lower mortality rates [65]. However, the evidence remains varied across studies [66]. Furthermore, VDD has been linked to the production and maintenance of pro-inflammatory state in obesity [67]. The strong association between VDD and AIDs, and other conditions characterized by an exacerbated inflammatory response, underscores the intricate interplay between vitamin D and the immune system. Future epidemiological research must include the assessment of cofounding variables related to VDD and diseases with exacerbated inflammatory response, especially lifestyle factors such as diet and exercise.

## 4. Vitamin D and Gallbladder Cancer

GBC represents a unique malignancy not only due to its distinct clinical presentation but also because the combination of its risk factors remains poorly understood [5]. Numerous risk factors for GBC have been identified, including gallstones, chronic diseases, female sex, geographical location, ethnicity, congenital developmental abnormalities, overweight/obesity, bacterial and parasitic infections, autoimmune diseases, genetic variants, nutritional aspects, smoking, and alcohol consumption [5,6].

The presence of gallstones is considered the primary risk factor for GBC [68]. Low-grade dysplasia is predominantly found in the fundus of the gallbladder, providing further support for the role of gallstones in GBC carcinogenesis [8]. Interestingly, GBC and gallstones share several risk factors, yet only a small proportion of individuals with gallstones go on to develop GBC [69]. For instance, the Chile Biliary Longitudinal Study reported a gallstone prevalence of 24.7% among Chilean women from high-risk locations, with an expected GBC incidence of approximately 2% during the 6-year follow-up period [68]. Genetic predisposition, including factors such as sex, geographical location, ethnicity, and genetic variants, has emerged as a significant risk factor [6].

However, genetic factors are non-modifiable, thus providing insight into high-risk populations through genetic studies aids in identifying individuals who may benefit from preventive measures. Nevertheless, it does not address the implementation of feasible preventive actions from a public health perspective. On the other hand, chronic inflammatory responses, both local and systemic, have been implicated as major drivers of GBC carcinogenesis, associated with chronic and infectious diseases, nutritional factors, and lifestyle choices [6]. Importantly, these factors are generally modifiable or treatable.

Regarding gallbladder diseases, VDD has been suggested as a risk factor for gallstones. However, a large population study found no association between VDD and gallstones, although an intriguing link was identified between sunlight/VDD exposure during fetal development and gallbladder stasis, which is a risk factor for gallstone formation [70]. VDD has also been associated with other gallbladder diseases such as gallbladder mucocele [71].

The association between VDR polymorphisms and cancer susceptibility has been extensively investigated across various malignancies, including prostate, breast, colorectal, and skin cancer [72]. Significant associations have been identified with specific VDR polymorphisms, such as Fok1, Bsm1, Taq1, and Apa1, suggesting that VDR genetic variation may modulate cancer risk [73,74]. However, conflicting data and a lack of consensus exist regarding the impact of VDR polymorphisms on cancer risk across different populations and cancer types. Factors such as ethnicity, phenotype, 25(OH)D plasma levels, and UV radiation exposure complicate these findings, indicating the need for a comprehensive assessment integrating genetic, environmental, and lifestyle factors [72]. Additionally, studies have explored the potential interactions between VDR polymorphisms and cancer-related phenotypes, including tumor aggressiveness, response to treatment, and clinical outcomes. For example, in prostate cancer, the VDR ApaI polymorphism was significantly associated with prostate-specific antigen levels and tumor stage, suggesting a role in disease progression and prognosis [73]. Haplotype analysis has revealed combinations of VDR polymorphisms associated with increased cancer risk, emphasizing the complex genetic architecture underlying cancer susceptibility [75]. These findings underscore the importance of considering VDR genetic variations alongside clinical parameters for a comprehensive understanding of cancer biology and personalized therapeutic interventions. In the context of biliary tract cancer and autoimmune diseases, VDR polymorphisms have been associated with an elevated risk of GBC and primary biliary cirrhosis [71,76].

Hence, VDD appears to favor gallstone development through cholestasis. Moreover, VDD has been shown to increase biliary fibrosis [76], while a VDR agonist has demonstrated the ability to reduce oxidative stress-induced senescence in bile duct epithelium [77]. Additionally, reduced VDR expression has been observed in GBC samples compared to benign conditions [78], and VDD has been associated with an incomplete response to ursodeoxycholic acid (UDCA) in primary biliary cirrhosis patients [79]. Recently, a cohort study involving cholangiocarcinoma patients emphasized that vitamin D intake enhances disease-free survival (DFS) in individuals undergoing surgery [80].

From an epidemiological perspective, future studies must assess the association between VDD and GBC (incidence and severity) in populations with a high incidence of GBC, as these populations could differ in the strength of this association.

## 5. Vitamin D Reduces the Th17/Treg Ratio and Suppresses Cancer Cells’ Invasiveness

The available evidence suggests that vitamin D plays a suppressive role in promoting tolerogenic responses, which explains its positive effects on autoimmune and inflammatory diseases [12,81]. This response involves effects on both innate and adaptive immune cells [82]. In monocytes and macrophages, vitamin D increases the phagocytic response and production of antimicrobial proteins, thereby enhancing the antimicrobial activity of the innate immune system. In dendritic cells (DCs), vitamin D downregulates the expression of antigen-presenting and costimulatory molecules, as well as pro-inflammatory cytokines, while upregulating inhibitory molecules, anti-inflammatory cytokines, and molecules associated with T cell polarization and Treg development [82].

Additionally, vitamin D is required to maintain the bioenergetic parameters of peripheral blood mononuclear cells (PBMCs). VDD in adults has been correlated with increased oxidative metabolism and activation of PBMCs [83]. The increase in Treg induced by vitamin D has been attributed to the reduction in methylation in the forkhead box protein 3 (FoxP3) promoter and the regulation of glucose metabolism in DCs [84,85]. On the other hand, the decrease in Th17 cells has been explained by the regulation of NF-κB activity and the inhibition of the expression of elated orphan receptor γt (RORγt/IL-17) in animal models [86]. Furthermore, patients with rheumatoid arthritis treated with Tocilizumab, an IL-6 receptor inhibitor, have shown a better response when their vitamin D levels were adequate, suggesting a suppression of IL-17 and Th17 cells [87]. The balance between Th17 and Treg cells has significant implications in autoimmune and inflammatory diseases [88].

In hepatopancreatobiliary diseases, damaged cholangiocytes can trigger a pro-inflammatory and profibrotic response. VDR ablation has been associated with an exacerbated pro-inflammatory response, while a vitamin D analog has been shown to reduce this response [89]. Vitamin D has also been found to prevent the activation of hepatic stellate cells and ameliorate inflammatory liver damage, although it does not appear to have an effect on fibrosis [90]. Therefore, the vitamin D-induced tolerogenic and anti-inflammatory response is related to a better response in autoimmune and inflammatory diseases, which may be mediated by a reduced Th17/Treg ratio. In CRC, the serum levels of 25(OH)D were observed to be downregulated in this cancer, suggesting an association between vitamin D and CRC pathogenesis. The diminished 25(OH)D levels in CRC patients were found to be linked with an augmented Th17/Treg ratio, as well as elevated levels of transforming growth factor (TGF)-β1, IL-10, IL-17, and IL-23 in peripheral blood. These findings suggest a potential association between VDD and altered immune responses in CRC, warranting further investigation into its immunomodulatory effects [91].

Vitamin D has been strongly associated with the suppression of cancer cell proliferation and the regulation of the tumor microenvironment to facilitate tumor repression [92]. In several types of cancer, vitamin D has been shown to suppress tumor growth and inhibit EMT [93,94]. Vitamin D analogs have also demonstrated the ability to reduce the apoptosis of PBMCs induced by PDAC cells [95]. Clinical evidence further supports the role of VDR activation in improving the response to chemotherapy and disease-free survival in hepatopancreatobiliary cancers [80]. Although vitamin D has been shown to suppress the proliferation of cancer cells through various mechanisms, including the inhibition of EMT, further research is needed to determine the specific role of Th17/Treg balance in mediating these effects. Nonetheless, the existing evidence suggests that the immune-modulating properties of vitamin D may contribute to its anti-cancer effects.

Tumor budding, which represents the presence of detached tumor cells with mesenchymal features at the tumor invasion front, is considered a manifestation of EMT [26]. It is associated with poor prognosis due to increased an invasiveness and poorer histological characteristics in GBC [24]. Additionally, there is a strong correlation between EMT and chemotherapy resistance in GBC, highlighting the significance of EMT in the carcinogenesis and prognosis of GBC [96]. Interestingly, studies have demonstrated the impact of vitamin D and IL-17A on EMT in different contexts. Vitamin D has been found to reduce EMT in intestinal fibrosis, suggesting its potential inhibitory effect on EMT-related processes [93]. Conversely, IL-17A has been shown to increase EMT and invasiveness in GBC [97]. Th17 cells can induce an inflammatory response associated with EMT, angiogenesis, and metastasis. This effect is mediated by the activation of signal transducer and transcription 3 (STAT3) and its interaction with cancer stem cells. However, Th17 cells can also suppress tumor growth through CD8+ T cell activation in an interferon-gamma (IFN-γ)-dependent manner [98]. Moreover, vitamin D is known to modulate various transcription factors that play critical roles in EMT regulation, such as Snail, Slug, and Twist [26]. By inhibiting the expression and activity of these transcription factors, vitamin D can potentially suppress EMT, thereby reducing tumor invasiveness and metastatic potential. Furthermore, vitamin D has been shown to enhance radiotherapy sensitivity and reduce chemotherapy resistance by inhibiting EMT in CRC patients [93,99].

In summary, EMT plays a crucial role in the invasiveness, metastasis, and chemotherapy resistance of GBC. The regulation of EMT involves complex interactions between various molecular factors, including vitamin D, IL-17A, and the Th17 immune response. Understanding these interactions and their implications for GBC carcinogenesis and prognosis may provide potential therapeutic avenues for intervention and improved patient outcomes. Future research must consider the assessment of vitamin D levels for patients undergoing a cholecystectomy, exploring the association between VDD and the presence of dysplastic changes (as EMT) in the biopsies. Again, high-incidence GBC populations are of particular interest for this future research.

## 6. The Influence of Th17/Treg Balance on the Immune Cells Landscape and Carcinogenesis

The shift in the balance between regulatory Treg and Th17 cells is pivotal in influencing disease progression. Recent discoveries have identified novel subsets, such as IL-17-producing Treg and IL-10-producing Th17 cells, highlighting their versatility and impact on various health conditions. These cells can be likened to the ancient concept of Yin and Yang, with immune-activating populations (e.g., Th17, M1 macrophages) representing Yang, and immunosuppressive populations (e.g., Treg, M2 macrophages) representing Yin. This analogy underscores the dynamic nature of immune responses, enriching our understanding of immune balance and its implications for health and disease [100].

Both excessive inflammation from Th17 cells and immunosuppression from Tregs play significant roles in the process of carcinogenesis. However, this balance varies depending on the type and stage of cancer [22], as well as the location (infiltrating or peripheral) of Th17/Treg cells [98]. The prevailing view suggests that the tolerogenic effects associated with Tregs lead to a pro-tumor response through changes in cytokine secretion, molecule expression, and alterations in T cell metabolism, ultimately inhibiting the anti-cancer immune response [101]. It has been observed that lactate in the tumor microenvironment (TME) enhances tryptophan metabolism and kynurenine production by DCs, contributing to the induction of Tregs [102]. Tregs have a metabolic advantage in low-glucose, lactate-rich environments, such as the TME [103]. Additionally, Tregs preferentially oxidize pyruvate in the mitochondria but can also utilize lactate as an alternative fuel source for oxidative phosphorylation (OXPHOS), which is essential for their suppressive functions [104]. Therefore, Treg cells in the TME during the advanced stages of cancer exert immunosuppressive effects that promote cancer progression.

On the other hand, Th17 cells rely on glycolysis for their differentiation. Despite being an inefficient source of ATP compared to OXPHOS, glycolysis fuels a PI3K-centered positive feedback regulatory circuit that drives effector responses [105]. Functionally, Th17 cells can induce an inflammatory response associated with EMT, angiogenesis, and metastasis through mechanisms involving STAT3 activation and interaction with cancer stem cells. They can also suppress tumors in an IFN-γ-dependent manner by activating CD8+ T cells [97,98]. Based on these findings, it has been proposed that blocking inflammation and regulating metabolic dysfunctions in the TME are necessary to improve cancer therapy, particularly for advanced stages [106].

There is a crucial role of RORγt+ cells, including type 3 innate lymphoid cells and Janus cells, in inducing Treg cells, with their absence leading to the proliferation of pathogenic Th17 cells. In CRC, an increased infiltration of CXCL13+ T cells, particularly Th17 cells, has been observed, alongside interactions between secreted phosphoprotein 1 (SPP1)+ tumor-associated macrophages (TAMs) and Treg cells. Additionally, CRC patients exhibit changes in Treg subpopulations influenced by the gut microbiota, indicating a complex interplay between immune cells and the microbiome in tumor progression [107,108,109]. In this context, Garcinia yunnanensis (YTE-17) shows promise in preventing CRC by modulating TME and immune cell dynamics. YTE-17 inhibits the Wingless-related integration site (Wnt)5a/Jun *N-*terminal kinase (JNK) signaling pathway, suppressing macrophage-mediated Th17 cell induction and restoring the balance between Treg and Th17 cells in colitis-associated cancer (CAC) models. This action reduces M2 macrophage polarization and Th17 cell phenotype, alongside inhibiting glycolysis in Th17 cells, highlighting YTE-17’s potential in mitigating CRC risk [110]. The differential expression of steroid receptor coactivator (SRC) in Tregs and Th17 cells highlights their distinct roles in immune regulation and tumor immunity, suggesting potential therapeutic targets for immune modulation in cancer treatment [111]. These findings emphasize the intricate interplay between immune cells and TME in colorectal and gastric cancers, offering promising avenues for targeted immunotherapies and prognostic biomarker development.

In breast cancer, γδ Treg cells induce senescence in DCs, suppressing Th1 and Th17 differentiation while promoting Treg development. Blocking PD-L1 and STAT3 pathways reverses these effects, enhancing tumor-specific immune responses. Similarly, cervical cancer studies highlight functional polarization of Th1, Th2, Th17, and Treg cells in responders, whereas non-responders exhibit increased programmed death-1 (PD-1) expression and dysfunctional tumor-infiltrating lymphocytes (TILs). In renal clear cell carcinoma, LINC00941 expression correlates positively with Tregs and Th2 cells but negatively with Th17 cells, impacting the immune landscape and prognosis [112,113,114]. In head and neck squamous cell carcinoma (HNSCC), increased IL-17-related molecules are linked to favorable prognosis, where mature DCs balance Treg and Th17 cells. In hepatocellular carcinoma (HCC), EEF1E1 expression is associated with various immune cells, including Tregs and Th17 cells, suggesting its potential as a prognostic marker. Additionally, in H. pylori-induced gastritis, IL-6 from DCs and macrophages induces IL-17A expression in FoxP3+ T cells, correlating with advanced lesions. These findings underscore the significant roles of Th17 and Treg cells in shaping the immune environment across different cancers, providing insights into potential therapeutic targets [115,116,117].

## 7. The Dichotomic Role of Th17/Treg Balance in GBC Carcinogenesis

Notably, certain conditions and deficits associated with exacerbated inflammation, such as primary sclerosing cholangitis, Crohn’s disease, and systemic lupus erythematosus, increase the risk of developing GBC, underscoring the role of inflammation in carcinogenesis [6]. Overall, the balance between Th17 and Treg cells is a crucial and dynamic process in carcinogenesis, involving context-dependent molecular, cellular, and metabolic pathways. Recent research has revealed a diverse array of T cell subtypes within TME, highlighting the complex interplay between immune cells and tumor progression. Among the identified subtypes, exhausted CD8+ T cells, activated/exhausted CD8+ T cells, and Tregs emerge as predominant players within GBC tumors. Notably, an inverse relationship between Th17 and Treg populations is observed, with Th17 levels notably reduced while Tregs exhibit a concomitant increase. This imbalance underscores the intricate immune dynamics characterizing GBC pathogenesis. Moreover, this study pioneers the establishment of subtyping criteria, delineating three distinct subtypes of GBC based on their pro-tumorigenic microenvironments. For instance, the type 1 subtype is characterized by heightened M2 macrophage infiltration, whereas the type 2 subtype exhibits a complex milieu comprising highly exhausted CD8+ T cells, B cells, and Tregs with suppressive activities. These findings provide valuable insights into the heterogeneity of T cell responses in GBC, suggesting that molecular subtyping based on T cells could offer a promising avenue for targeted immunotherapeutic strategies aimed at improving GBC treatment outcomes [118].

In GBC, like other cancers, the infiltration of immune cells serves as an independent prognostic indicator. A higher infiltration of FoxP3+ Tregs or CD15+ cells and the expression of inhibitory molecules are associated with unfavorable outcomes in GBC patients [119,120,121,122]. Conversely, an increased infiltration of CD8+ T cells and the absence of FoxP3+ Tregs correlate with prolonged survival, particularly in advanced stages [120,121,122,123]. Therefore, the paradigm that assumes the tolerogenic effects of Tregs lead to a pro-tumor response appears to hold true in GBC as well. However, it is important to note that the distribution of immune cells infiltrating the tumor core versus the tumor border may differ. Kim et al. [120] observed differences in the spatial distribution of immune cells in BTCs, including GBC. They found a higher density of CD8+ cytotoxic T cells, FoxP3− CD4+ helper T cells, and FoxP3+ CD4+ Tregs at the tumor margin. The density of lymphocyte activation gene 3 (LAG3)+/T-cell immunoglobulin and mucin domain 3 (TIM3)+ Tregs was similar in the tumor center and margin, and it was not associated with survival. Finally, a high density of FoxP3− CD4+ helper T cells in the tumor margin was independently associated with favorable progression-free survival (PFS) and overall survival (OS).

More recently, single-cell RNA sequencing (scRNA-seq) analysis revealed spatial heterogeneity in GBC, highlighting the immunosuppressive and tumor-promoting characteristics of immune cells within the GBC TME. Notably, CD4+/FoxP3+ Tregs and CD4+/CXCL13+ T helper cells with a higher expression of exhaustion biomarkers were found within GBC [124]. In GBC liver metastases, the presence of exhausted CD8+ T cells and immunosuppressive Tregs was prominent [125]. While most studies support the association between high-density infiltrating Tregs and poor survival in the advanced stages of GBC, they also underscore the high heterogeneity of the GBC TME in terms of cancer cells, stromal cells, and immune cells.

Interestingly, Kinoshita et al. [126] found that, during cancer progression, Tregs accumulate in the center of the tumor, whereas Th17 cells localize at the tumor front. Both patterns were indicative of poor prognosis. Furthermore, this pattern was associated with the secretion of IL-6 and TGF-β1 by cancer cells, with IL-6 inducing Th17 cells and TGF-β1 inducing Tregs, and the IL-6/Th17 axis was associated with EMT features. Interestingly, the association between IL-6 and EMT has been observed in various cancers [127]. Additionally, IL-6 has been identified as an independent predictor for type 2 diabetes mellitus (DM2), and GBC patients with DM2 exhibit higher mortality rates [128]. GBC patients also have elevated serum levels of IL-6, and increased Tγδ17, Th17, and Treg cells in peripheral blood have been associated with poor survival [129].

However, assessments of this heterogeneity during carcinogenesis are limited, and the link between exacerbated inflammation associated with various diseases and deficits and GBC progression remains incompletely understood. Given the role of inflammation in GBC carcinogenesis, investigating the Th17/Treg balance/heterogeneity and EMT throughout the GBC TME during carcinogenesis warrants attention, particularly in the early stages when interventions are typically more effective and yield better outcomes.

A recent investigation underscores the significant spatial reprogramming of regulatory and immunosuppressive cells, such as myeloid-derived suppressor cells (MDSCs), TAMs, Treg, and exhausted T cells (Tex), throughout the invasive and expansive phases of CRC. Particularly noteworthy is the association of these cells with invasion-related genes, notably the C-X-C motif ligand 1 (CXCL1) and CXCL8 genes, which bear implications for CRC prognosis. In stage I, EMT program-adjacent cells predominantly foster an inflammatory margin-invasive niche, while in stage III tissues, they are more likely to facilitate a hypoxic pre-invasive niche [130]. This indirect evidence complicates the interaction between cancer and immune cells, not only in terms of space but also in terms of time.

Furthermore, it is important to note that the imbalance in the Th17/Treg axis is not definitively established, as the upregulation of the Treg population in cancer is associated with increased metastasis. For instance, studies have shown that the administration of exogenous Treg into melanoma tumor-bearing mice leads to a significant increase in lung metastasis [131]. Additionally, in HCC, infiltrating Treg cells have been implicated in triggering the TGFβ1 signaling pathway and promoting the EMT of cancer cells during tumor hematogenous dissemination, potentially increasing the invasive potential of HCC cells [132]. Moreover, macrophages, NK cells, and Treg were found to be significantly enriched in the high-risk group of HCC patients [133]. Moreover, in HNSCC, SCC7/C3H syngeneic mouse models, the combination of anti-PD-1 antibody with melatonin significantly inhibited tumor growth and modulated anti-tumor immunity by increasing CD8+ T cell infiltration and decreasing the proportion of Treg in the TME [134]. These findings challenge the hypothesis linking VDD and the Th17/Treg axis, as the Treg-increasing effect of vitamin D could potentially enhance tumor growth and invasiveness.

However, additional evidence supports the protective function of Treg in TME. In PDAC, a heightened presence of CD3+CD8- T cells, primarily CD4+ T cells expressing FoxP3 (Tregs), has been linked to enhanced patient survival. Interestingly, the infiltration of cytotoxic CD3+CD8+ T cells has not shown a significant impact on overall survival. Transcriptomic analysis of PDAC tumors has identified three distinct signatures related to EMT/stromal, metabolic, and secretory/pancreatic processes. However, none of these signatures have explained the variance in Treg infiltration. Consequently, Tregs are associated with improved overall survival in PDAC patients, regardless of cytotoxic T cell infiltration or tumor transcriptomic profiles [135]. Furthermore, an analysis of data from the Cancer Genome Atlas (TCGA) has revealed that high-Treg CRCs exhibit enrichment in several pro-cancer signaling pathways compared to low-Treg CRCs, including EMT, Kras, Hypoxia, TGF-β, TNF-α, and angiogenesis. Surprisingly, Treg infiltration has shown a correlation with the earlier stages of CRC in TCGA data, and a higher proportion of Tregs has been associated with an improved response to chemotherapy [136]. Indeed, there is conflicting evidence regarding the role of Treg in carcinogenesis.

Additionally, there are confounding variables and alternative interpretations for the observed associations between VDD, immune modulation, and cancer progression. Lifestyle habits, genetic predispositions, and environmental factors could influence both VDD and GBC risk. However, a critical caveat for this hypothesis is that VDS did not lead to a reduction in cancer mortality in the primary meta-analysis of randomized clinical trials (RCTs), as the observed 6% risk reduction was not statistically significant. However, in a post hoc analysis focusing on trials with daily dosing, adults aged 70 years and older, as well as individuals who started VDS before their cancer diagnosis, appeared to derive the greatest benefit from daily VDS [137].

Future research assessing the association between VDD and dysplastic changes (as EMT) in cholecystectomy biopsies must include the characterization of tumor-infiltrating T cells in tumor border and tumor core for patients with confirmed GBC, and how the spatial distribution of Th17/Treg ratio is associated with EMT features across the tumor. On the other hand, for cholecystectomy biopsies without GBC, the same assessment must focus on areas of the gallbladder with exacerbated inflammation. By examining these regions, researchers can elucidate the impact of VDD on immune cell infiltration and inflammatory responses in the absence of malignancy. This comparative approach will help distinguish the specific contributions of VDD to tumor development and inflammatory processes within the gallbladder microenvironment.

## 8. Final Remarks

Based on these data, we hypothesize that vitamin D contributes to EMT regulation in GBC by modulating the spatial distribution of tumor-infiltrating T cells and particularly by modulating the Th17/Treg balance in the tumor border. Consequently, the Th17/Treg imbalance may serve as a plausible mediator linking VDD to early GBC carcinogenesis. The triad VDD-Th17/Treg-EMT could be implicated in GBC carcinogenesis and prognosis, as depicted in Figure 1.

GBC represents a multifaceted disease influenced by a myriad of etiological factors, encompassing chronic inflammatory responses stemming from persistent infections, autoimmune conditions, and diverse lifestyle determinants. VDD has been extensively investigated in the context of AID, where an imbalance between Th17 and Treg has been identified as a contributing factor. Similarly, numerous epidemiological studies have established an association between VDD and various types of cancer, including GBC. The effects of vitamin D are mediated through VDR and have been demonstrated in multiple solid organ cancers.

TME plays a critical role in cancer development and progression. While the presence of a Th17/Treg imbalance has been observed in the TME, the distribution of immune cells within the TME exhibits heterogeneity, leading to conflicting findings across studies. To gain a more comprehensive understanding of the relationship between VDD, the Th17/Treg balance, and carcinogenesis, future investigations incorporate both ex vivo evaluations in patients with GBC and cholelithiasis and in vitro evaluations of the impact of vitamin D on the Th17/Treg balance/heterogeneity and EMT in cholangiocytes. Currently, research in this field often focuses solely on either in vitro or ex vivo experiments, limiting our ability to fully comprehend the complex processes underlying carcinogenesis at both the patient and cellular/molecular levels.

It is worth noting that while this hypothesis specifically focuses on GBC, VDD has been associated with various other types of solid organ cancers. Therefore, elucidating the role of the Th17/Treg imbalance as a potential mediator between VDD and EMT in GBC would not only provide valuable insights into GBC pathogenesis but also have implications for understanding the development of other solid organ cancers, given the emerging significance of Th17/Treg cells in cancer research.

However, GBC carcinogenesis is an intricate process and some limitations must be considered to evaluate the hypothesis. In the gallbladder, VDR is predominantly activated by vitamin D. However, the role of vitamin D in GBC carcinogenesis becomes more complex because BA can also activate VDR [138]. Additionally, BA can activate retinoic acid-related orphan receptor γt (RORγt), which is associated with Th17 cells [2]. Furthermore, enzymes involved in BA metabolism and vitamin D metabolism are related to GBC [139], and a chronic BA infection has also been implicated in GBC [140]. These interactions appear to be secondary to the immunomodulatory effects exerted by vitamin D. Overall, VDD seems to be associated with several gallbladder diseases, including GBC, but the underlying mechanisms of this association are yet to be fully elucidated.

There is a pressing need for more rigorous evidence and mechanistic studies to establish a definitive causal link between VDD and cancer. Although significant associations have been observed in observational and ecological studies, these alone cannot establish causality due to potential confounding factors and the complex nature of cancer etiology. To elucidate the biological mechanisms through which VDD may influence GBC carcinogenesis, targeted mechanistic studies are essential. Recent experimental studies and clinical trials, which investigate the role of vitamin D in cellular processes such as proliferation, apoptosis, and immune modulation, are beginning to offer a more detailed and nuanced understanding of the potential causal relationship between VDD and GBC.

This hypothesis not only aims to enhance our understanding of the complex immunological mechanisms involved in GBC initiation but also holds significant clinical implications. It suggests the potential benefit of incorporating VDS into preventative strategies for high-risk populations, such as individuals with gallstones. By addressing the elevated incidence and mortality rates associated with VDD in GBC, this approach could contribute to more effective preventive measures and potentially reduce the burden of this malignancy.

## Figures and Tables

**Figure 1 nutrients-16-04134-f001:**
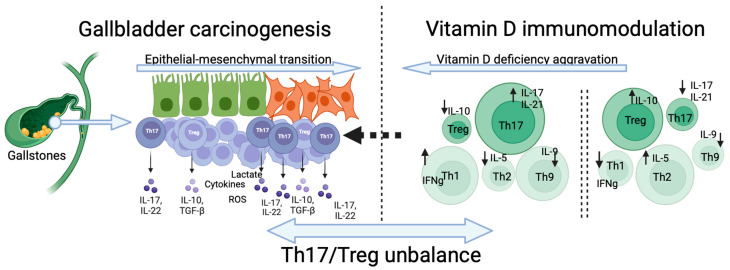
Immunomodulatory role of vitamin D in restoring Th17/Treg balance and suppressing epithelial–mesenchymal transition in gallbladder cancer.

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
