# Peer review of "The Immunomodulatory Role of Vitamin D in Regulating the Th17/Treg Balance and Epithelial–Mesenchymal Transition: A Hypothesis for Gallbladder Cancer"

_nutrients, 2024, doi:10.3390/nu16234134_

Round 1
Reviewer 1 Report (New Reviewer)
Comments and Suggestions for Authors
The authors in the paper investigated the influence of Vitamin D deficiency on Gall Bladder Cancer with a focus on immunological pathways. In their study the authors included lots of case studies and schematic mechanism for better understanding the interplay between various Vitamin D receptor and immune response which is the strength of the paper. Additionally, the authors highlighted the crucial role of Th17/Treg balance and epithelial-mesenchymal transition in the invasiveness, metastasis, and chemotherapy resistance of cancer. In conclusion, the authors have supported their hypothesis with clinical studies and provided a further direction to implement the outcomes. This paper can be accepted for the publication.
Author Response
The authors in the paper investigated the influence of Vitamin D deficiency on Gall Bladder Cancer with a focus on immunological pathways. In their study the authors included lots of case studies and schematic mechanism for better understanding the interplay between various Vitamin D receptor and immune response which is the strength of the paper. Additionally, the authors highlighted the crucial role of Th17/Treg balance and epithelial-mesenchymal transition in the invasiveness, metastasis, and chemotherapy resistance of cancer. In conclusion, the authors have supported their hypothesis with clinical studies and provided a further direction to implement the outcomes. This paper can be accepted for the publication.
Response: Thank you for your comment.
Reviewer 2 Report (Previous Reviewer 2)
Comments and Suggestions for Authors
In the revised version, the authors addressed most of my comments and concerns regarding the manuscript.
Author Response
In the revised version, the authors addressed most of my comments and concerns regarding the manuscript.
Response: Thank you for your comment.
Reviewer 3 Report (New Reviewer)
Comments and Suggestions for Authors
This is a comprehensive and detailed catalogue of published information relevant to understanding possible roles of vitamin D deficiency in the etiology of gallbladder cancer. It is difficult to follow the logic of many sections of the manuscript because each sentence is simply describing some published finding, followed by another sentence with further published findings. The section (lines157 to 191) dealing with genes possibly linked to susceptibility for gallbladder cancer, is particularly difficult to follow because of the many unexplained acronyms, such as BTC, FGF, IDH, PBKCA, HERs, TP53, KRAS, ALK, EGFR and BRAF. Because so many acronyms are used throughout the text, perhaps it would help a reader to identify the points being made if there was a table of acronyms with an explanation as to the relevance of these to the topic of gallbladder cancer.
There are some minor points that need to be addressed:
1. Line 65: “ergocalciferol, which is derived from plants”. This is incorrect. Ergocalciferol is produced by ultraviolet irradiation of ergosterol, a sterol in fungi. If ergosterol is present in plants, it is because of endophytic fungi growing on those plants which are then exposed to solar UV light.
2. Lines 85-89: A definition of vitamin D deficiency is needed here. Vitamin D inadequacy is defined as “serum concentration < 50nmol/L”. This should state that it is the serum concentration of 25-hydroxyvitamin D and should define the lower limit of inadequacy below which vitamin D deficiency would be the diagnosis.
3. Line 93: “GBC cancer” – the word “cancer is redundant as the term would mean: Gallbladder cancer cancer.
4. Some acronyms remain a mystery. Line 282, “AID” appears to be undefined. Line 385, “EMT” appears to be undefined.
5. The second sentence in the title has the words: “A hypotheses” . The word Hypotheses is plural. Perhaps that sentence should read: “An hypothesis in Gallbladder cancer”
Author Response
This is a comprehensive and detailed catalogue of published information relevant to understanding possible roles of vitamin D deficiency in the etiology of gallbladder cancer. It is difficult to follow the logic of many sections of the manuscript because each sentence is simply describing some published finding, followed by another sentence with further published findings. The section (lines157 to 191) dealing with genes possibly linked to susceptibility for gallbladder cancer, is particularly difficult to follow because of the many unexplained acronyms, such as BTC, FGF, IDH, PBKCA, HERs, TP53, KRAS, ALK, EGFR and BRAF. Because so many acronyms are used throughout the text, perhaps it would help a reader to identify the points being made if there was a table of acronyms with an explanation as to the relevance of these to the topic of gallbladder cancer.
Response: Thank you for your constructive feedback. To enhance readability, we have added a comprehensive list of acronyms as Section #9, providing explanations for each term.
There are some minor points that need to be addressed:
1. Line 65: “ergocalciferol, which is derived from plants”. This is incorrect. Ergocalciferol is produced by ultraviolet irradiation of ergosterol, a sterol in fungi. If ergosterol is present in plants, it is because of endophytic fungi growing on those plants which are then exposed to solar UV light.
Response: We appreciate this correction and have revised the text to clarify that ergocalciferol is produced through ultraviolet irradiation of ergosterol, a sterol found in fungi. As noted, any presence of ergosterol in plants results from endophytic fungi exposed to UV light.
2. Lines 85-89: A definition of vitamin D deficiency is needed here. Vitamin D inadequacy is defined as “serum concentration < 50nmol/L”. This should state that it is the serum concentration of 25-hydroxyvitamin D and should define the lower limit of inadequacy below which vitamin D deficiency would be the diagnosis.
Response: We have revised the definition for clarity. Based on Amrein et al. (2020), we now define vitamin D deficiency as a serum 25-hydroxyvitamin D [25(OH)D] concentration of <50 nmol/L (20 ng/mL), and we have removed the term "vitamin D inadequacy" for simplicity and considering that VDI is usually considered a mild form of VDD.
3. Line 93: “GBC cancer” – the word “cancer is redundant as the term would mean: Gallbladder cancer cancer.
Response: We have removed the redundant term as suggested.
4. Some acronyms remain a mystery. Line 282, “AID” appears to be undefined. Line 385, “EMT” appears to be undefined.
Response: We have clarified these terms in the new list of acronyms provided in Section #9.
5. The second sentence in the title has the words: “A hypotheses” . The word Hypotheses is plural. Perhaps that sentence should read: “An hypothesis in Gallbladder cancer”
Response: We have corrected the title to "A hypothesis in gallbladder cancer."
Round 2
Reviewer 3 Report (New Reviewer)
Comments and Suggestions for Authors
The modifications to the manuscript do make it easier to understand. However, the discussion section is very dense although with diligent concentration is can be understood. The list of acronyms is a great help.
This manuscript is a resubmission of an earlier submission. The following is a list of the peer review reports and author responses from that submission.
Round 1
Reviewer 1 Report
Comments and Suggestions for Authors
While the hypothesis presented in the article is intriguing and the potential link between Vitamin D deficiency (VDD), T-helper 17 (Th17)/regulatory T (Treg) imbalance, and gallbladder cancer (GBC) development is a compelling area of research, there are several critical issues and gaps in the rationale and design of this review that limit its current impact.
First, the review posits a direct relationship between VDD, Th17/Treg imbalance, and epithelial-mesenchymal transition (EMT) in GBC based on associations seen in autoimmune disorders and other solid organ cancers. However, it does not adequately address the complexity and heterogeneity of GBC, potentially oversimplifying the pathway from VDD to cancer progression. GBC is influenced by a multitude of risk factors and molecular pathways, and the review does not sufficiently contextualize the proposed mechanism within this broader landscape.
Moreover, the review appears to rely heavily on extrapolations from studies in autoimmunity and other cancers without providing direct evidence from GBC research. This raises questions about the specificity of the proposed mechanism to GBC and whether similar effects of vitamin D on Th17/Treg balance and EMT have been observed in this context. The leap from observed phenomena in one domain to another without direct evidence weakens the argument and undermines the review's credibility.
Additionally, while the immunomodulatory effects of vitamin D and its impact on Th17/Treg balance are well-documented, the review does not critically assess the evidence or consider contradictory findings that might challenge the proposed hypothesis. A more balanced and comprehensive review of the literature, including studies that may not support the hypothesis, would provide a stronger foundation for the arguments presented.
The review also does not adequately address potential confounding factors and alternative explanations for the associations observed between VDD, immune modulation, and cancer progression. For example, lifestyle factors, genetic predispositions, and environmental exposures that might influence both VDD and GBC risk are not thoroughly considered. This omission leaves the review vulnerable to criticism for not accounting for these critical variables in its analysis.
Finally, while the review proposes investigating the role of Th17/Treg imbalance as a mediator between VDD and EMT in GBC, it does not offer a clear methodology or approach for such an investigation. Given the complexity of the immune response in cancer and the challenges of studying these processes in vivo, a more detailed discussion of potential experimental designs, model systems, and analytical techniques would strengthen the review and provide a clearer path forward for research in this area.
In summary, while the hypothesis linking VDD, Th17/Treg imbalance, and GBC development through EMT modulation presents a novel avenue for research, the review as currently presented lacks the necessary depth, critical analysis, and methodological detail to convincingly support this theory. Further incorporation of direct evidence from GBC studies, and a more comprehensive review of the literature are needed to make a compelling case for the proposed research direction.
Comments on the Quality of English LanguageWhile the hypothesis presented in the article is intriguing and the potential link between Vitamin D deficiency (VDD), T-helper 17 (Th17)/regulatory T (Treg) imbalance, and gallbladder cancer (GBC) development is a compelling area of research, there are several critical issues and gaps in the rationale and design of this review that limit its current impact.
First, the review posits a direct relationship between VDD, Th17/Treg imbalance, and epithelial-mesenchymal transition (EMT) in GBC based on associations seen in autoimmune disorders and other solid organ cancers. However, it does not adequately address the complexity and heterogeneity of GBC, potentially oversimplifying the pathway from VDD to cancer progression. GBC is influenced by a multitude of risk factors and molecular pathways, and the review does not sufficiently contextualize the proposed mechanism within this broader landscape.
Moreover, the review appears to rely heavily on extrapolations from studies in autoimmunity and other cancers without providing direct evidence from GBC research. This raises questions about the specificity of the proposed mechanism to GBC and whether similar effects of vitamin D on Th17/Treg balance and EMT have been observed in this context. The leap from observed phenomena in one domain to another without direct evidence weakens the argument and undermines the review's credibility.
Additionally, while the immunomodulatory effects of vitamin D and its impact on Th17/Treg balance are well-documented, the review does not critically assess the evidence or consider contradictory findings that might challenge the proposed hypothesis. A more balanced and comprehensive review of the literature, including studies that may not support the hypothesis, would provide a stronger foundation for the arguments presented.
The review also does not adequately address potential confounding factors and alternative explanations for the associations observed between VDD, immune modulation, and cancer progression. For example, lifestyle factors, genetic predispositions, and environmental exposures that might influence both VDD and GBC risk are not thoroughly considered. This omission leaves the review vulnerable to criticism for not accounting for these critical variables in its analysis.
Finally, while the review proposes investigating the role of Th17/Treg imbalance as a mediator between VDD and EMT in GBC, it does not offer a clear methodology or approach for such an investigation. Given the complexity of the immune response in cancer and the challenges of studying these processes in vivo, a more detailed discussion of potential experimental designs, model systems, and analytical techniques would strengthen the review and provide a clearer path forward for research in this area.
In summary, while the hypothesis linking VDD, Th17/Treg imbalance, and GBC development through EMT modulation presents a novel avenue for research, the review as currently presented lacks the necessary depth, critical analysis, and methodological detail to convincingly support this theory. Further incorporation of direct evidence from GBC studies, and a more comprehensive review of the literature are needed to make a compelling case for the proposed research direction.
Author Response
Thank you for your thorough review and insightful comments. We appreciate the opportunity to address each of your concerns and have made revisions accordingly.
While the hypothesis presented in the article is intriguing and the potential link between Vitamin D deficiency (VDD), T-helper 17 (Th17)/regulatory T (Treg) imbalance, and gallbladder cancer (GBC) development is a compelling area of research, there are several critical issues and gaps in the rationale and design of this review that limit its current impact.
First, the review posits a direct relationship between VDD, Th17/Treg imbalance, and epithelial-mesenchymal transition (EMT) in GBC based on associations seen in autoimmune disorders and other solid organ cancers. However, it does not adequately address the complexity and heterogeneity of GBC, potentially oversimplifying the pathway from VDD to cancer progression. GBC is influenced by a multitude of risk factors and molecular pathways, and the review does not sufficiently contextualize the proposed mechanism within this broader landscape.
Response: Thank you for your comment. We appreciate your acknowledgment of the complexity inherent in GBC carcinogenesis. Indeed, while numerous risk factors and molecular pathways have been identified, the conclusive evidence regarding their interplay remains elusive due to a lack of comprehensive understanding of the cellular and molecular mechanisms involved in this specific carcinogenesis process. In our review, we aimed to provide a critical analysis of both direct and indirect evidence linking VDD to GBC progression, drawing insights from analogous pathways observed in other cancers and autoimmune diseases. We agree that GBC represents a multifactorial disease influenced by a variety of factors, and our intention was not to oversimplify its pathogenesis but rather to offer a focused exploration of the potential immunological mechanisms involved. In response to your feedback, we have refined the aim of our paper to explicitly acknowledge this complexity. The revised aim now states: “to critically assess the existing literature concerning the influence of VDD on GBC and the immunological pathways linked to its carcinogenesis based on evidence from cancer biology and autoimmune diseases.” Furthermore, the paper was re-entitled to: “The Immunomodulatory Role of Vitamin D in Regulating Th17/Treg Balance and Epithelial-Mesenchymal Transition. A Hypotheses in Gallbladder Cancer.” We believe that this clarification better reflects our intention to provide a nuanced understanding of the interplay between vitamin D, immune dysregulation, and GBC progression within the broader landscape of cancer biology and autoimmune pathogenesis.
Moreover, the review appears to rely heavily on extrapolations from studies in autoimmunity and other cancers without providing direct evidence from GBC research. This raises questions about the specificity of the proposed mechanism to GBC and whether similar effects of vitamin D on Th17/Treg balance and EMT have been observed in this context. The leap from observed phenomena in one domain to another without direct evidence weakens the argument and undermines the review's credibility.
Response: Thank you for your feedback. As mentioned, we have included both direct and indirect evidence in our review. While we acknowledge that extrapolating evidence from other cancers and autoimmune diseases to a specific cancer like GBC may seem like a significant leap, our intention is not merely to summarize existing evidence but rather to propose a hypothesis for further investigation. We understand the concerns raised about the specificity of the proposed mechanism to GBC and the need for direct evidence in this context. Our aim is to stimulate further research in this area and provide a foundation for future studies to test the hypothesis we have proposed.
Additionally, while the immunomodulatory effects of vitamin D and its impact on Th17/Treg balance are well-documented, the review does not critically assess the evidence or consider contradictory findings that might challenge the proposed hypothesis. A more balanced and comprehensive review of the literature, including studies that may not support the hypothesis, would provide a stronger foundation for the arguments presented.
Response: Thank you for your valuable feedback. We appreciate your insightful comments regarding the need for a more balanced and comprehensive review of the literature. We acknowledge the importance of critically assessing the evidence and considering contradictory findings that may challenge the proposed hypothesis. In response to your comment, we have included some contradictory findings in the section titled "Final Remarks" at the end of the paper, where we address the limitations of our review. This section highlights the complexity of carcinogenesis and immunological pathways related to vitamin D in gallbladder cancer, and emphasizes the need for further research to fully elucidate these mechanisms. We hope that these additions enhance the depth and credibility of our review, and we appreciate your thoughtful suggestions for improving the paper.
The review also does not adequately address potential confounding factors and alternative explanations for the associations observed between VDD, immune modulation, and cancer progression. For example, lifestyle factors, genetic predispositions, and environmental exposures that might influence both VDD and GBC risk are not thoroughly considered. This omission leaves the review vulnerable to criticism for not accounting for these critical variables in its analysis.
Response: Thank you for your comment. We completely agree, and we added this comment as a limitation.
Finally, while the review proposes investigating the role of Th17/Treg imbalance as a mediator between VDD and EMT in GBC, it does not offer a clear methodology or approach for such an investigation. Given the complexity of the immune response in cancer and the challenges of studying these processes in vivo, a more detailed discussion of potential experimental designs, model systems, and analytical techniques would strengthen the review and provide a clearer path forward for research in this area.
Response: Thank you for your valuable feedback. We appreciate your suggestion regarding the inclusion of a clear methodology or approach for investigating the role of Th17/Treg imbalance as a mediator between VDD and EMT in GBC. While we agree that discussing potential experimental designs, model systems, and analytical techniques would enhance the review, we believe that such detailed methodology is more suited for a research proposal or grant application rather than a review manuscript. However, we recognize the importance of providing guidance for future research directions. In response to your comment, we plan to develop a separate research proposal outlining the methodology and approach for testing our hypothesis. This will allow us to delve into the specifics of experimental designs and analytical techniques in a more appropriate format.
In summary, while the hypothesis linking VDD, Th17/Treg imbalance, and GBC development through EMT modulation presents a novel avenue for research, the review as currently presented lacks the necessary depth, critical analysis, and methodological detail to convincingly support this theory. Further incorporation of direct evidence from GBC studies, and a more comprehensive review of the literature are needed to make a compelling case for the proposed research direction.
Reviewer 2 Report
Comments and Suggestions for Authors
In this manuscript, authors hypothesized that vitamin D inhibited EMT through regulating Th17/Treg ration in gallbladder cancer (GBC). They also summarized the immunomodulation of vitamin D, as well as the Th17/Treg balance could impact on carcinogenesis of GBC. These literatures were integrated to provide an overview of the connections between immunomodulation of vitamin D and EMT in various solid cancers. In my opinion, some major revision should be resolved.
1. Title and abstract should be improved.
2. As a humoral factor, 1a,25(OH)2D3, avidly binds and activated VDR. Alteration in both VDR expressions and polymorphisms, and in the synthesis (25-hydroxylase and 1a hydroxylase) and catabolism (24-hydrocylase) of vitamin D metabolites could compromise 1a,25(OH)2D3 sensitivity and 1a,25(OH)2D3 signaling. Please provide literatures for the association of synthesis and metabolism of vitamin D with GBC.
3. The figure 1 needs to be better structured, such as both EMT and Th17/Treg unbalance promote carcinogenesis of GBC, vitamin D including VDR and its metabolic enzymes modulates the EMT and Th17/Treg ratio.
Author Response
Thank you for your thorough review and insightful comments. We appreciate the opportunity to address each of your concerns and have made revisions accordingly.
In this manuscript, authors hypothesized that vitamin D inhibited EMT through regulating Th17/Treg ration in gallbladder cancer (GBC). They also summarized the immunomodulation of vitamin D, as well as the Th17/Treg balance could impact on carcinogenesis of GBC. These literatures were integrated to provide an overview of the connections between immunomodulation of vitamin D and EMT in various solid cancers. In my opinion, some major revision should be resolved.
1. Title and abstract should be improved.
Response: We have modified the title to “The Immunomodulatory Role of Vitamin D in Regulating Th17/Treg Balance and Epithelial-Mesenchymal Transition. A Hypotheses in Gallbladder Cancer.” Abstract was re-written in order to improve clarity.
2. As a humoral factor, 1a,25(OH)2D3, avidly binds and activated VDR. Alteration in both VDR expressions and polymorphisms, and in the synthesis (25-hydroxylase and 1a hydroxylase) and catabolism (24-hydrocylase) of vitamin D metabolites could compromise 1a,25(OH)2D3 sensitivity and 1a,25(OH)2D3 signaling. Please provide literatures for the association of synthesis and metabolism of vitamin D with GBC.
Response: We found no direct evidence on the association of synthesis and metabolism of vitamin D with GBC.
3. The figure 1 needs to be better structured, such as both EMT and Th17/Treg unbalance promote carcinogenesis of GBC, vitamin D including VDR and its metabolic enzymes modulates the EMT and Th17/Treg ratio.
Direct evidence linking the synthesis and metabolism of vitamin D to gallbladder cancer (GBC) remains elusive based on our review. While some evidence exists regarding polymorphisms of the vitamin D receptor (VDR) and its potential association with GBC, it does not align with the hypothesis outlined in Figure 1.
Round 2
Reviewer 1 Report
Comments and Suggestions for Authors
This review article presents a hypothesis on the potential role of vitamin D deficiency (VDD) in influencing gallbladder cancer (GBC) progression through the modulation of T-helper 17 (Th17) and regulatory T (Treg) cell balance. There are several critical shortcomings that could undermine the impact of the findings.
1. The manuscript discusses associative studies between VDD and cancer but lacks a strong presentation of evidence or mechanistic studies that well establish a causal link. Additional comprehensive discussion on the strength of the current evidence and its limitations would enhance the review’s depth.
2. The review hypothesizes a specific effect of vitamin D on the Th17/Treg balance in GBC but does not sufficiently address how these effects are uniquely relevant to GBC compared to other cancers. A more detailed explanation of why GBC is particularly susceptible to these immunomodulatory effects would strengthen the argument.
3. Any contradictory findings or ongoing debates within the field regarding the role of vitamin D in cancer and autoimmunity should be addressed. This would provide a more balanced view and acknowledge the complexity of the research landscape.
4. The manuscript should discuss how the Th17/Treg balance interacts with other immune cells in the tumor microenvironment, such as macrophages, natural killer cells, and dendritic cells. Understanding these interactions could reveal additional insights into how VDD affects immune responses and cancer progression.
5. The review should consider the potential impact of vitamin D receptor (VDR) polymorphisms on GBC risk and progression. This could include genetic studies exploring associations between VDR variants and GBC incidence or outcomes, providing a genetic basis for the relationship between VDD and cancer.
6. A discussion on how VDD influences other molecular pathways involved in cancer progression, such as WNT/β-catenine or JAK/STAT signaling, would provide a broader view of its role in carcinogenesis. This could include exploring how these pathways interact with the Th17/Treg balance.
7. A discussion on how VDD impacts the broader tumor microenvironment in GBC is needed. This includes its influence on stromal cells, extracellular matrix remodeling, and angiogenesis, providing a comprehensive view of its effects.
8. The review could benefit from a meta-analysis of existing studies on VDD and GBC, providing a statistical synthesis of available evidence. This would help clarify the relationship between VDD, immune responses, and cancer progression.
9. The review should delve deeper into the role of vitamin D metabolism in influencing cancer risk, particularly the enzymatic pathways that convert vitamin D into its active form. This could include discussing variations in these pathways and their impact on vitamin D levels and GBC progression.
10.The review should explore how genetic factors, such as polymorphisms in genes related to vitamin D metabolism or immune regulation, influence the relationship between VDD and GBC. This could include studies assessing gene-diet interactions and their impact on cancer risk.
11.A discussion on inflammation-related biomarkers, such as C-reactive protein (CRP) or cytokine levels, would provide insights into how inflammation influences GBC progression and how vitamin D supplementation might modulate these markers.
12.The review should consider the impact of vitamin D supplementation on other autoimmune conditions beyond GBC, providing a broader context for its immunomodulatory effects. This could include discussing how these conditions influence cancer risk or progression.
By addressing these additional points, the manuscript would provide a more comprehensive and clinically relevant contribution to GBC research, offering valuable insights into the relationship between VDD and cancer progression and its potential therapeutic implications.
Comments on the Quality of English LanguageThis review article presents a hypothesis on the potential role of vitamin D deficiency (VDD) in influencing gallbladder cancer (GBC) progression through the modulation of T-helper 17 (Th17) and regulatory T (Treg) cell balance. There are several critical shortcomings that could undermine the impact of the findings.
1. The manuscript discusses associative studies between VDD and cancer but lacks a strong presentation of evidence or mechanistic studies that well establish a causal link. Additional comprehensive discussion on the strength of the current evidence and its limitations would enhance the review’s depth.
2. The review hypothesizes a specific effect of vitamin D on the Th17/Treg balance in GBC but does not sufficiently address how these effects are uniquely relevant to GBC compared to other cancers. A more detailed explanation of why GBC is particularly susceptible to these immunomodulatory effects would strengthen the argument.
3. Any contradictory findings or ongoing debates within the field regarding the role of vitamin D in cancer and autoimmunity should be addressed. This would provide a more balanced view and acknowledge the complexity of the research landscape.
4. The manuscript should discuss how the Th17/Treg balance interacts with other immune cells in the tumor microenvironment, such as macrophages, natural killer cells, and dendritic cells. Understanding these interactions could reveal additional insights into how VDD affects immune responses and cancer progression.
5. The review should consider the potential impact of vitamin D receptor (VDR) polymorphisms on GBC risk and progression. This could include genetic studies exploring associations between VDR variants and GBC incidence or outcomes, providing a genetic basis for the relationship between VDD and cancer.
6. A discussion on how VDD influences other molecular pathways involved in cancer progression, such as WNT/β-catenine or JAK/STAT signaling, would provide a broader view of its role in carcinogenesis. This could include exploring how these pathways interact with the Th17/Treg balance.
7. A discussion on how VDD impacts the broader tumor microenvironment in GBC is needed. This includes its influence on stromal cells, extracellular matrix remodeling, and angiogenesis, providing a comprehensive view of its effects.
8. The review could benefit from a meta-analysis of existing studies on VDD and GBC, providing a statistical synthesis of available evidence. This would help clarify the relationship between VDD, immune responses, and cancer progression.
9. The review should delve deeper into the role of vitamin D metabolism in influencing cancer risk, particularly the enzymatic pathways that convert vitamin D into its active form. This could include discussing variations in these pathways and their impact on vitamin D levels and GBC progression.
10.The review should explore how genetic factors, such as polymorphisms in genes related to vitamin D metabolism or immune regulation, influence the relationship between VDD and GBC. This could include studies assessing gene-diet interactions and their impact on cancer risk.
11.A discussion on inflammation-related biomarkers, such as C-reactive protein (CRP) or cytokine levels, would provide insights into how inflammation influences GBC progression and how vitamin D supplementation might modulate these markers.
12.The review should consider the impact of vitamin D supplementation on other autoimmune conditions beyond GBC, providing a broader context for its immunomodulatory effects. This could include discussing how these conditions influence cancer risk or progression.
By addressing these additional points, the manuscript would provide a more comprehensive and clinically relevant contribution to GBC research, offering valuable insights into the relationship between VDD and cancer progression and its potential therapeutic implications.
Author Response
1. The manuscript discusses associative studies between VDD and cancer but lacks a strong presentation of evidence or mechanistic studies that well establish a causal link. Additional comprehensive discussion on the strength of the current evidence and its limitations would enhance the review’s depth.
Thank you for your insightful comment. We have added further comments to the Discussion section to provide a more comprehensive analysis of the current evidence, including its strengths and limitations.
2. The review hypothesizes a specific effect of vitamin D on the Th17/Treg balance in GBC but does not sufficiently address how these effects are uniquely relevant to GBC compared to other cancers. A more detailed explanation of why GBC is particularly susceptible to these immunomodulatory effects would strengthen the argument.
We appreciate this observation. Given the multifactorial nature of carcinogenesis, our focus on GBC stems from existing epidemiological evidence linking VDD to GBC. However, this link is not exclusive to GBC but is also seen in other solid organ cancers. For the purpose of this paper, we have explored GBC as part of a grant proposal. Therefore, we do not imply that GBC is uniquely susceptible to these immunomodulatory effects. This context has now been clarified in the manuscript.
3. Any contradictory findings or ongoing debates within the field regarding the role of vitamin D in cancer and autoimmunity should be addressed. This would provide a more balanced view and acknowledge the complexity of the research landscape.
We have expanded our discussion on the contradictory findings in cancer research, particularly concerning the Th17 and Treg ratio, to provide a more balanced perspective on the ongoing debates in the field.
4. The manuscript should discuss how the Th17/Treg balance interacts with other immune cells in the tumor microenvironment, such as macrophages, natural killer cells, and dendritic cells. Understanding these interactions could reveal additional insights into how VDD affects immune responses and cancer progression.
Thank you for this suggestion. We have included a new section that discusses the interactions between Th17/Treg cells and other immune cells within the tumor microenvironment.
5. The review should consider the potential impact of vitamin D receptor (VDR) polymorphisms on GBC risk and progression. This could include genetic studies exploring associations between VDR variants and GBC incidence or outcomes, providing a genetic basis for the relationship between VDD and cancer.
We have incorporated relevant data from the literature regarding the association of VDR polymorphisms with various cancers, including GBC.
6. A discussion on how VDD influences other molecular pathways involved in cancer progression, such as WNT/β-catenin or JAK/STAT signaling, would provide a broader view of its role in carcinogenesis. This could include exploring how these pathways interact with the Th17/Treg balance.
We have integrated a discussion on how VDD influences other molecular pathways involved in cancer progression, particularly focusing on the interaction of Th17/Treg cells with these pathways.
7. A discussion on how VDD impacts the broader tumor microenvironment in GBC is needed. This includes its influence on stromal cells, extracellular matrix remodeling, and angiogenesis, providing a comprehensive view of its effects.
While we recognize the importance of these factors, this discussion falls outside the scope of our current manuscript. However, we have added a brief comment in the relevant section acknowledging the broader impacts of vitamin D on various cell types.
8. The review could benefit from a meta-analysis of existing studies on VDD and GBC, providing a statistical synthesis of available evidence. This would help clarify the relationship between VDD, immune responses, and cancer progression.
Conducting a meta-analysis is beyond the scope of our current paper. Nonetheless, we have highlighted the existing epidemiological evidence linking VDD with GBC and acknowledged that similar associations exist for other solid organ cancers.
9. The review should delve deeper into the role of vitamin D metabolism in influencing cancer risk, particularly the enzymatic pathways that convert vitamin D into its active form. This could include discussing variations in these pathways and their impact on vitamin D levels and GBC progression.
We have not found direct evidence linking the enzymatic pathways that convert vitamin D into its active form with GBC progression. Therefore, this discussion has not been included.
10. The review should explore how genetic factors, such as polymorphisms in genes related to vitamin D metabolism or immune regulation, influence the relationship between VDD and GBC. This could include studies assessing gene-diet interactions and their impact on cancer risk.
The discussion on polymorphisms related to vitamin D metabolism and their association with various cancers is already included. However, we have found no direct evidence on gene-diet interactions specifically affecting GBC risk.
11. A discussion on inflammation-related biomarkers, such as C-reactive protein (CRP) or cytokine levels, would provide insights into how inflammation influences GBC progression and how vitamin D supplementation might modulate these markers.
We have incorporated a brief discussion on the relationship between vitamin D levels (and supplementation) and C-reactive protein. The association with cytokine levels is already addressed in another section of our paper.
12. The review should consider the impact of vitamin D supplementation on other autoimmune conditions beyond GBC, providing a broader context for its immunomodulatory effects. This could include discussing how these conditions influence cancer risk or progression.
While this is indeed an important area, it is outside the scope of our current paper. We have acknowledged the extensive evidence showing the negative impact of VDD on autoimmune diseases, which is already mentioned in our manuscript. We have added more evidence on the effects of vitamin D supplementation on cancers.
By addressing these additional points, the manuscript provides a more comprehensive and clinically relevant contribution to GBC research, offering valuable insights into the relationship between VDD and cancer progression and its potential therapeutic implications.
Reviewer 2 Report
Comments and Suggestions for Authors
In the revised manuscript, authors did some work to improve manuscript.
Round 3
Reviewer 1 Report
Comments and Suggestions for Authors
The revised manuscript nutrients-2959700 R2 investigates the relationship between vitamin D deficiency (VDD) and the pathogenesis of gallbladder cancer (GBC), positing that VDD influences cancer development through an imbalance in T-helper 17 (Th17) and regulatory T (Treg) cells and affects the epithelial-mesenchymal transition (EMT). Despite its some interesting approach to examining immunomodulation in cancer, the study still exhibits substantial experimental design flaws that necessitate major revisions or reconsideration for publication.
1. The manuscript broadly links VDD with chronic inflammation and cancer progression but fails to provide a direct evidence connecting VDD specifically to GBC. The hypothesis lacks specificity as the direct effects of VDD on GBC carcinogenesis are not well-established through existing literature should be provided in introduction and discussion section by the authors.
2. The introduction section does not account for other significant risk factors for GBC such as genetic predispositions, gallstones, or lifestyle factors beyond VDD. A more comprehensive analysis considering these confounders is necessary to isolate the effect of Vitamin D on GBC development.
3. The manuscript lacks specificity in directly linking the immunological pathways discussed to GBC. Cancer-specific immunological environments can vary significantly, and the generalization from broader cancer or autoimmune contexts to GBC weakens the manuscript's impact. A focused examination of how these pathways are uniquely altered in GBC is necessary.
4. The study does not adequately address the need for control experiments, particularly negative and positive controls that could validate the specific effects of vitamin D on Th17/Treg balance and EMT in GBC. Without these controls, the specificity of the observed effects cannot be confidently attributed to vitamin D.
5. While the manuscript hypothesizes a connection between vitamin D, Th17/Treg balance, and EMT in gallbladder cancer, it lacks in-depth mechanistic studies that could clarify these pathways at the molecular level. Detailed studies, possibly involving molecular docking, signaling pathway analysis, or gene expression profiling, would substantiate the mechanisms.
6. The potential long-term effects of vitamin D modulation in gallbladder cancer patients are not discussed. Long-term follow-up studies would be essential to understand the sustained impacts of vitamin D on cancer progression, recurrence, and patient survival.
In conclusion, while the revised manuscript nutrients-2959700 R2 presents a hypothesis linking Vitamin D deficiency with gallbladder cancer progression through immune modulation and EMT, it falls short in providing the necessary depth and original data to advance this theory convincingly. The authors must address these major revision comments by incorporating more data to bridge the gaps between Vitamin D, immune response modulation, and cancer pathogenesis. Without these improvements, the manuscript does not currently meet the standards for publication in a Nutrients (ISSN 2072-6643) journal.
Comments on the Quality of English LanguageThe revised manuscript nutrients-2959700 R2 investigates the relationship between vitamin D deficiency (VDD) and the pathogenesis of gallbladder cancer (GBC), positing that VDD influences cancer development through an imbalance in T-helper 17 (Th17) and regulatory T (Treg) cells and affects the epithelial-mesenchymal transition (EMT). Despite its some interesting approach to examining immunomodulation in cancer, the study still exhibits substantial experimental design flaws that necessitate major revisions or reconsideration for publication.
1. The manuscript broadly links VDD with chronic inflammation and cancer progression but fails to provide a direct evidence connecting VDD specifically to GBC. The hypothesis lacks specificity as the direct effects of VDD on GBC carcinogenesis are not well-established through existing literature should be provided in introduction and discussion section by the authors.
2. The introduction section does not account for other significant risk factors for GBC such as genetic predispositions, gallstones, or lifestyle factors beyond VDD. A more comprehensive analysis considering these confounders is necessary to isolate the effect of Vitamin D on GBC development.
3. The manuscript lacks specificity in directly linking the immunological pathways discussed to GBC. Cancer-specific immunological environments can vary significantly, and the generalization from broader cancer or autoimmune contexts to GBC weakens the manuscript's impact. A focused examination of how these pathways are uniquely altered in GBC is necessary.
4. The study does not adequately address the need for control experiments, particularly negative and positive controls that could validate the specific effects of vitamin D on Th17/Treg balance and EMT in GBC. Without these controls, the specificity of the observed effects cannot be confidently attributed to vitamin D.
5. While the manuscript hypothesizes a connection between vitamin D, Th17/Treg balance, and EMT in gallbladder cancer, it lacks in-depth mechanistic studies that could clarify these pathways at the molecular level. Detailed studies, possibly involving molecular docking, signaling pathway analysis, or gene expression profiling, would substantiate the mechanisms.
6. The potential long-term effects of vitamin D modulation in gallbladder cancer patients are not discussed. Long-term follow-up studies would be essential to understand the sustained impacts of vitamin D on cancer progression, recurrence, and patient survival.
In conclusion, while the revised manuscript nutrients-2959700 R2 presents a hypothesis linking Vitamin D deficiency with gallbladder cancer progression through immune modulation and EMT, it falls short in providing the necessary depth and original data to advance this theory convincingly. The authors must address these major revision comments by incorporating more data to bridge the gaps between Vitamin D, immune response modulation, and cancer pathogenesis. Without these improvements, the manuscript does not currently meet the standards for publication in a Nutrients (ISSN 2072-6643) journal.
Author Response
The revised manuscript nutrients-2959700 R2 investigates the relationship between vitamin D deficiency (VDD) and the pathogenesis of gallbladder cancer (GBC), positing that VDD influences cancer development through an imbalance in T-helper 17 (Th17) and regulatory T (Treg) cells and affects the epithelial-mesenchymal transition (EMT). Despite its some interesting approach to examining immunomodulation in cancer, the study still exhibits substantial experimental design flaws that necessitate major revisions or reconsideration for publication.
R: Thank you for recognizing the potential interest in our approach. In response to your feedback, we have made several significant revisions to the manuscript to address the concerns raised. We trust these changes will enhance the clarity and rigor of our study.
1. The manuscript broadly links VDD with chronic inflammation and cancer progression but fails to provide a direct evidence connecting VDD specifically to GBC. The hypothesis lacks specificity as the direct effects of VDD on GBC carcinogenesis are not well-established through existing literature should be provided in introduction and discussion section by the authors.
R: Our previous responses in this matter have been pointing out that there is a relationship between VDD and gallbladder pathologies, most of the evidence comes from epidemiological studies. However, we also pointed out that VDD is correlated with several solid organ cancers, thus we do not claim specificity in this matter, moreover in the final section of our review we explicit mention that or hypothesis could add relevant evidence for another solid organ cancers.
2. The introduction section does not account for other significant risk factors for GBC such as genetic predispositions, gallstones, or lifestyle factors beyond VDD. A more comprehensive analysis considering these confounders is necessary to isolate the effect of Vitamin D on GBC development.
R: We have included a new entire section on the risk factors for GBC.
3. The manuscript lacks specificity in directly linking the immunological pathways discussed to GBC. Cancer-specific immunological environments can vary significantly, and the generalization from broader cancer or autoimmune contexts to GBC weakens the manuscript's impact. A focused examination of how these pathways are uniquely altered in GBC is necessary.
R: We appreciate the reviewer’s insightful comments. As previously mentioned, our manuscript emphasizes the association between VDD and GBC, with much of the supporting evidence derived from epidemiological studies. We acknowledge that VDD has been linked to various solid organ cancers, and thus, we do not assert a cancer-specific specificity in this context. In the concluding section of our review, we clearly state that our hypothesis may contribute valuable insights not only for GBC but also for other cancer types. Regarding the immunological pathways specific to GBC, while conclusive evidence is limited, we have cited relevant studies by Chen, Kinoshita, Zhang, Jing, and others, which are discussed in the section titled "The Dichotomic Role of Th17/Treg Balance in GBC Carcinogenesis." These studies, though not definitive, offer important contributions to the understanding of immunological mechanisms in GBC.
4. The study does not adequately address the need for control experiments, particularly negative and positive controls that could validate the specific effects of vitamin D on Th17/Treg balance and EMT in GBC. Without these controls, the specificity of the observed effects cannot be confidently attributed to vitamin D.
R: We acknowledge the importance of incorporating control experiments to validate the specific effects of VDD on Th17/Treg balance and EMT in GBC. To address this concern, we have elaborated on our research design, including comprehensive plans for both negative and positive controls, in the supplementary file that accompanies our submission. This file, which is not intended for publication, provides detailed information on the methodologies we will employ to ensure the specificity and reliability of our findings. We appreciate your feedback and are committed to enhancing the rigor of our study through these control measures.
5. While the manuscript hypothesizes a connection between vitamin D, Th17/Treg balance, and EMT in gallbladder cancer, it lacks in-depth mechanistic studies that could clarify these pathways at the molecular level. Detailed studies, possibly involving molecular docking, signaling pathway analysis, or gene expression profiling, would substantiate the mechanisms.
R: While the manuscript proposes a hypothesis linking VDD, Th17/Treg balance, and EMT in GBC, it indeed lacks comprehensive mechanistic studies to elucidate these pathways at the molecular level. We acknowledge that detailed investigations such as molecular docking, signaling pathway analysis, and gene expression profiling would provide substantial insights into these mechanisms. However, addressing these aspects falls beyond the scope of this review and the current research project slated to commence next year. We appreciate the suggestion and agree that future studies incorporating these methodologies could significantly enhance our understanding of the underlying mechanisms involved.
6. The potential long-term effects of vitamin D modulation in gallbladder cancer patients are not discussed. Long-term follow-up studies would be essential to understand the sustained impacts of vitamin D on cancer progression, recurrence, and patient survival.
R: While direct evidence on the long-term effects of VDS in GBC patients is currently lacking, we have included pertinent data from studies involving elderly, adult, and pediatric populations. This broader context provides valuable insights that may offer indirect implications for understanding the potential long-term impacts of vitamin D on cancer progression, recurrence, and patient survival.
Round 4
Reviewer 1 Report
Comments and Suggestions for Authors
The authors have thoroughly addressed all my previous concerns, and I appreciate their efforts in revising the manuscript (Manuscript No.: nutrients-2959700 R1). Their comprehensive response has resolved all potential issues. Consequently, I believe the manuscript is now suitable for publication. However, I recommend a professional English proofreading before final submission to correct any remaining typographical errors or minor oversights.
Comments on the Quality of English LanguageThe authors have thoroughly addressed all my previous concerns, and I appreciate their efforts in revising the manuscript (Manuscript No.: nutrients-2959700 R1). Their comprehensive response has resolved all potential issues. Consequently, I believe the manuscript is now suitable for publication. However, I recommend a professional English proofreading before final submission to correct any remaining typographical errors or minor oversights.